# Comparative Study of Biochar Modified with Different Functional Groups for Efficient Removal of Pb(II) and Ni(II)

**DOI:** 10.3390/ijerph191811163

**Published:** 2022-09-06

**Authors:** Chengcheng Liu, Jiaxin Lin, Haojia Chen, Wanjun Wang, Yan Yang

**Affiliations:** 1School of Environmental and Safety Engineering, Changzhou University, Changzhou 213164, China; 2School of Environmental Science and Engineering, Institute of Environmental Health and Pollution Control, Guangdong University of Technology, Guangzhou 510006, China; 3Synergy Innovation Institute of GDUT, Shantou 515041, China; 4Chemistry and Chemical Engineering Guangdong Laboratory, Shantou 515041, China

**Keywords:** adsorption, modified biochar, corn stalk, Ni(II), Pb(II)

## Abstract

The potential application of biochar in water treatment is attracting interest due to its sustainability and low production cost. In the present study, H_3_PO_4_-modified porous biochar (H-PBC), ethylenediaminetetraacetic acid-modified porous biochar (E-PBC), and NaOH-modified porous biochar (O-PBC) were prepared for Ni(II) and Pb(II) adsorption in an aqueous solution. Scanning electron microscopy (SEM), X-ray diffraction analysis (XRD), Brunauer–Emmett–Teller analysis (BET), and Fourier-transform infrared (FT-IR) spectroscopy were employed to characterize the as-obtained samples, and their capacities for Ni(II) and Pb(II) adsorption were determined. SEM showed that H-PBC retained the hierarchical porous structure of pristine biochar. FT-IR showed that H-PBC possessed abundant oxygen-containing and phosphorus-containing functional groups on the surface. BET analysis demonstrated that the surface areas of H-PBC (344.17 m^2^/g) was higher than O-PBC (3.66 m^2^/g), and E-PBC (1.64 m^2^/g), respectively. H-PBC, E-PBC, and O-PBC all exhibited excellent performance at Ni(II) and Pb(II) adsorption with maximum adsorption capacity of 64.94 mg/g, 47.17 mg/g, and 60.24 mg/g, and 243.90 mg/g, 156.25 mg/g, and 192.31 mg/g, respectively, which were significantly higher than the adsorption capacity (19.80 mg/g and 38.31 mg/g) of porous biochar (PBC). Pseudo-second order models suggested that the adsorption process was controlled by chemical adsorption. After three regeneration cycles, the Ni(II) and Pb(II) removal efficiency with H-PBC were still 49.8% and 56.3%. The results obtained in this study suggest that H-PBC is a promising adsorbent for the removal of heavy metals from aqueous solutions.

## 1. Introduction

The pollution of aquatic environments with heavy metals due to the natural and anthropogenic activity may seriously threaten human health. The problem of heavy metal pollution has become increasingly severe due to the rapid development of industrialization in recent decades [1]. Among the polluting heavy metals, Ni(II) is highly toxic and potentially carcinogenic, where it can damage the brain, spine, and internal organs of aquatic organisms and humans. Due to the adverse effects of Ni(II) on human health, it is classified as a human carcinogen by institutions around the world [2]. Pb(II) is also extremely harmful to the human body, where it can cause liver and kidney damage, or nervous system disorders by interacting with the sulfhydryl groups of proteins [3]. Combined pollution of water bodies with Ni(II) and Pb(II) can negatively impact the environment and human health. Therefore, there is an urgent need to develop effective methods for removing both Ni and Pb from the environment.

Various techniques have been developed for heavy metal removal, such as adsorption, ion exchange, electro-coagulation, and chemical precipitation [4,5,6,7]. Among these methods, adsorption using carbon materials has received increasing attention [8]. For instance, Mohammadi et al., [2] prepared activated carbon with a high surface area using ZnCl_2_ as a porogen, and the maximum adsorption capacities for Pb(II) and Ni(II) were 200 and 166.7 mg/g, respectively. Terracciano et al., [9] used graphene oxide for Ca(II) adsorption and found that graphene oxide had a high capacity for Ca(II) adsorption when the pH was greater than 7.0. However, these materials have relatively high costs and they involve complicated fabrication procedures. For instance, the preparation of graphene requires concentrated H_2_SO_4_ and HNO_3_ for exfoliation. Recent studies have shown that biochars obtained from waste biomass, such as rice husk [10] and peanut shells [11], have great potential as cost-effective adsorbents for Ni and Pb remediation in soil and water. Biochar has a high capacity for adsorbing heavy metals because of its porous structure and inhomogeneous surface chemistry [12].

However, biochar is normally produced by pyrolysis at high temperature, which inevitably reduces the number of functional groups on the biochar’s surface, thereby greatly decreasing the amount of active binding sites available for heavy metal adsorption. Thus, various methods have been applied to modify biochar by introducing different functional groups to enhance the capacity for adsorption. For example, Jiang et al., [8] synthesized H_3_PO_4_ modified biochar using a solvothermal method and the maximum Pb(II) adsorption capacity was 353 mg/g, thereby demonstrating its potential use in Pb-contaminated soil remediation. Choudhary et al., [13] modified biochar with NaOH to load more oxygen-containing functional groups on its surface and the capacity for Ni(II) removal was enhanced. In addition, ethylenediaminetetraacetic acid (EDTA) was used to modify biochar and the maximum Pb(II) adsorption capacity of the EDTA-functionalized biochar was 129.31 mg/g [4]. These previous studies demonstrated that modifying biochar with H_3_PO_4_, NaOH, and EDTA can improve the adsorption of Pb(II) and Ni(II), but the biochars used in these studies were derived from different raw materials under different synthesis conditions. The adsorption capacity of biochar is highly dependent on its physicochemical properties, which are determined by the different raw materials and synthesis conditions employed [14,15,16]. It is currently unclear which biochar modification technique is best for enhancing the removal of both Pb(II) and Ni(II). The effects on the heavy metal adsorption performance of the presence of different functional groups on modified biochar also require further exploration.

In this study, three representative modification methods were applied to synthesize biochar modified with H_3_PO_4_, NaOH, and EDTA using the same raw materials. The capacities of the newly synthesized biochars for adsorbing both Pb(II) and Ni(II) were systematically investigated in terms of their adsorption kinetics and thermodynamics. The results obtained in this study provide insights into the effects of different functional groups on biochar on the adsorption of Pb(II) and Ni(II), thereby facilitating the development of cost-effective modification strategies for synthesizing biochar for highly efficient heavy metal removal.

## 2. Experimental

### 2.1. Materials and Reagents

Lead acetate ((CH3COO)_2_Pb), Nickel sulfate (NiSO_4_), Potassium bicarbonate (KHCO_3_), Phosphoric acid (H_3_PO_4_), ethylenediaminetetraacetic acid (EDTA), Hydrochloric acid (HCl), and Sodium hydroxide (NaOH), were of analytical grade, obtained from Shanghai Anpu experimental technology Co. Ltd, Shanghai, China.

### 2.2. Synthesis of Biochars with Different Functional Groups

All of the biochar materials were synthesized using corn stalks as the raw materials, which were collected from farmland in Guiyu Town, Shantou City, China. The corn stalk biomass was washed with deionized water and dried naturally for 2 days before drying at 70 °C in an oven for more than 24 h until constant weight. The dried corn stalks were crushed using a ball mill (JX-2G) and passed through a 100-mesh sieve to obtain a constant particle size of less than 150 μm.

Porous biochar (PBC) was synthesized by pyrolyzing the corn stalk powder with added KHCO_3_ [17]. Briefly, the corn stalk powder and KHCO_3_ were mixed at a mass ratio of 1:1 and pyrolyzed in a tube furnace with a heating rate of 5 °C min^−1^ under N_2_ purging (200 mL/min) at 800 °C for 3 h, before washing with HCl (0.1 mol/L) to remove the ash. After subsequently washing with deionized water, the PBC product was dried in an oven for further use.

The H_3_PO_4_-modified porous PBC (designated as H-PBC) was prepared as described by Jiang et al., [8]. First, 5 g of PBC was mixed with 50 mL of H_3_PO_4_ solution and soaked for 30 min. The mixture was then transferred to a 100 mL polytetrafluoroethylene lined autoclave (50 ML, Changzhou, China) and heated at 240 °C for 2 h. After cooling to room temperature, the solid product was washed several times with deionized water and then dried overnight in an oven.

The NaOH-modified porous PBC (designated as O-PBC) was prepared as described by Choudhary et al., [13]. First, 5 g of NaOH was dissolved in 200 mL of deionized water and 5 g of PBC was added. The mixed solution was then stirred for 2 h and dried. The dried O-PBC was calcined at 500 °C at a heating rate of 10 °C/min for 1 h. Finally, the product was washed with deionized water and dried in oven overnight.

The EDTA-modified porous PBC (designated as E-PBC) was prepared as described by Li et al., [4]. First, 5 g PBC was mixed with 20 mL of aqueous acetic acid solution and 6.0 g of EDTA was mixed with 100 mL of methanol. The two mixtures were then combined and stirred for 24 h. The solid product was washed with deionized water and dried overnight.

The microscopic surface textures of the samples were examined by field emission scanning electron microscopy (S-3400N, S-570, Tokyo, Japan). The crystal structure of each biochar was examined by X-ray diffraction (D8 Advance, Tokyo, Japan) analysis at 45 kV and with 100 mA CuKα radiation. Fourier-transform infrared spectroscopy (Nicolet FTIR 6700, New York, NY, USA) was conducted to characterize the functional groups on the as-obtained biochar samples. The specific surface area of each test material was measured using Brunauer–Emmett–Teller analysis (3H-2000PS2, Beijing, China).

### 2.3. Pb and Ni Adsorption Experiments

The heavy metal adsorption experiments were conducted in a laboratory scale reactor. Specific amounts of biochar (5–50 mg) were mixed with containing 20–100 mL Ni(II)/Pb(II) solution (40 mg/L). The pH of the reaction mixture was adjusted to 2.0–9.0 with 0.1 M NaOH/HCl. Subsequently, adsorption was performed by shaking the reactor in a water bath where the temperature was maintained at 288.15–318.15 K. At specific time intervals (3, 6, 9, 12, and 15 h), samples were removed and centrifuged (SC-3610, Changzhou, China) at 3000 rpm for 10 min. The supernatant obtained was used to determine the Ni(II) and Pb(II) concentrations with a flame atomic absorption spectrometer (SP-3530AA, Shanghai, China).

The adsorption capacity *q_e_* (mg/g) and removal efficiency R was calculated for Ni(II) and Pb(II) using Equation (1) [18]:(1)$$q_{e}=V\left(C_{o}-C_{e}\right)/m$$
(2)$$R=\frac{C_{0}-C}{C_{0}}$$
where *C_o_* and *C_e_* are the heavy metal concentrations (mg/L) at the initial time and adsorption time, respectively, *V* is the volume (L) of the reaction solution, and *m* is the dried weight (g) of biochar. All of the adsorption experiments were conducted in triplicate.

### 2.4. Adsorption Kinetics and Isotherm Models

First-order and second-order models were used to study the adsorption kinetics with Equations (3) and (4) [18], respectively:(3)$$q_{t}=q_{e}\left(1-e^{-k_{1}t}\right)$$
(4)$$q_{t}=\frac{k_{2}q_{e}^{2}t}{1+k_{2}q_{e}t}$$
where *q_t_* (mg/g) is the adsorption capacity at time *t* (min), *q_e_* (mg/g) is the adsorption capacity at equilibrium, *k*_1_ (min^–1^) is the first order adsorption rate constant, and *k*_2_ [g/(mg·min)] is the second order adsorption rate constant.

Langmuir and Freundlich models were used to analyze the adsorption isotherms for Ni(II) and Pb(II) with different initial concentrations (10–100 mg/L). The equations for the Langmuir and Freundlich models are given in Equations (5) and (6) [18]:(5)$$\mathrm{Langmuir}:\;q_{e}=\frac{q_{m}K_{L}C_{e}\;}{1+K_{L}C_{e}\;}$$
(6)$$\mathrm{Freundlich}:\;q_{e}=K_{F}C_{e}{}^{1/n}$$
where *q_e_* (mg/g) is the adsorption capacity at equilibrium, *q_m_* (mg/g) is the maximum adsorption capacity, *C_e_* (mg/L) is the equilibrium concentration, *K_L_* (mg/L) is the Langmuir constant related to the adsorption energy, and *K_F_* and *n* are the Freundlich constant and nonlinear factor, respectively.

### 2.5. Recyclability Analysis

Five consecutive adsorption–desorption experiments were conducted in order to explore the reusability of the as-prepared modified biochar samples. First, 0.2 g of biochar was placed into a reactor containing 200 mL of Ni(NO_3_)_2_/Pb(NO_3_)_2_ (40 mg/L) solution. The suspension was stirred for 9 h to reach adsorption equilibrium. The used adsorbent was then collected by filtration, washed with ultrapure water, and re-used in the next run for Ni(II) and Pb(II) desorption. After five consecutive adsorption-desorption experiments, the adsorbent was withdrawn from the treated solution and regenerated by 0.5 mol.L^−1^ of NaOH solution, followed by washing with ultrapure water, 0.05 mol.L^−1^ HCl, and again ultrapure water, and then vacuum dried at 80 °C for reuse.

## 3. Results and Discussion

### 3.1. Characteristics of Adsorbents

The morphologies of the as-prepared samples were studied based on SEM images. Figure 1A shows that the PBC had a hierarchical porous structure similar to that of carbon foam. Large amounts of chelating agents were attached to the surface of E-PBC, which resulted in significant aggregation (Figure 1B). Rectangular crystals spread all over the surface of PBC in O-PBC (Figure 1C). The hierarchical porous structure of PBC was retained in H-PBC and projections were observed on its surface (Figure 1D). The existence of the porous structure agreed well with the result obtained by BET analysis.

Figure 2A shows the N_2_ adsorption–desorption isotherms determined for the as-prepared samples. The adsorption–desorption curves for PBC and H-PBC were type IV adsorption/desorption isotherms with H4 type hysteresis loops, and they suggested the existence of a mesoporous structure according to a previous study [17]. By contrast, the curves for E-PBC and O-PBC were type I adsorption/desorption isotherms due to their microporous filling. As shown in Figure 2B, most of the pores in PBC and H-PBC were mesopores. The BET surface areas determined for PBC, H-PBC, O-PBC, and E-PBC were 360.41, 344.17, 3.66, and 1.64 m^2^/g, respectively (Table 1). Compared with PBC, the surface area was clearly much lower for E-PBC, possibly because the micropores were covered by EDTA. The surface area of O-PBC was also significantly reduced. The reason is similar to EPBC. Interestingly, H-PBC maintained a high surface area in a similar manner to PBC, thereby indicating that modification with H_3_PO_4_ retained the porous structure of the raw material, as well as maintaining the high surface area of the biochar through the formation of mesopores.

The element analysis of the samples is shown in Table 1. It can be found that E-PBC and O-PBC have less C content than PBC, while H-PBC has more C content than PBC. The O content and H content showed a similar order. For H-PBC, the presence of P content proved that H_3_PO_4_ successfully binds to biochar. The H/C atomic ratio shows the aromaticity of a biochar due to hydrogenation and dehydrogenation process [18]. For H-PBC, E-PBC, and O-PBC, the H/C atom ratio decreased indicated that the samples were more aromatic and stable. The (N+O)/C ratio is related to the hydrophobicity and polarity of biochar [19]. For E-PBC, and O-PBC, the (N+O)/C atom ratio decreased revealed that increase in hydrophobicity of biochar after EDTA and NaOH modification, while H-PBC showed an opposite order.

The XRD patterns obtained for the PBC samples are illustrated in Figure 2C. For PBC, the characteristic peaks at 29.81°, and 43° corresponded to the (112), and (101) planes, respectively [17], which confirmed that PBC sample exhibited certain degree of graphitization. For H-PBC, the peak observed at 23° corresponded to the (002) planes of the graphite lattice [20] and the sharp peaks at 27.2°, 33.7°, and 35.2° were other diffraction peaks on H-PBC. According to the literature [4,9,14], amorphous materials do have high heavy metals adsorption capacity and as a result of H-PBC has excellent potential ability to remove heavy metals from aqueous solution. For O-PBC, a weak peak around 48° was related to (100) diffraction and another weak peak at around 43° was related to (101) diffraction by the activated biochar [21]. For E-PBC, the presence of new peaks at 24.72° confirming the reaction between PBC and EDTA to form amorphous polymer composite [22].

FTIR analysis was conducted to identify the functional groups on the surfaces of the prepared PBC samples and the results are shown in Figure 2D. For PBC, the adsorption peak around 3430 cm^–1^ could be attributed to the stretching of -NH groups in acidic and aliphatic compounds. The peak at 1049 cm^–1^ was due to the stretching vibration of C-O [17]. For O-PBC, the signals detected at 3430 cm^–1^ and 1442 cm^–1^ corresponded to the stretching vibrations of -OH and chromene, thereby, suggesting that O-PBC was successfully synthesized and the abundant functional groups may be beneficial to the adsorption of heavy metals. For H-PBC, the peaks at 1223 cm^–1^, 1185 cm^–1^, and 500 cm^–1^ could be assigned to the stretching vibrations of C-O-C, P-O-C [13], and C-P bonds [23], respectively. Those functional groups are attributed to a long-range chain order owing to the strong hydrogen bonding and polar intra- and inter-molecular interactions [14], which might strengthen binding to heavy metals. In addition, the bands at 1610 cm^–1^ and 833 cm^–1^ could be assigned to the stretching vibrations of aromatic C=C and =C-H, respectively [14], thereby indicating the high aromaticity of H-PBC. For E-PBC, the absorption peaks near 2925 cm^–1^, 1627 cm^–1^, and 1390 cm^–1^ could be attributed to the asymmetric stretching of -CH_2_- [24], C=O [22], and -NH_2_, thereby indicating that EDTA was successfully grafted onto PBC.

These results indicate that the E-PBC, O-PBC, and H-PBC were successfully synthesized according to previous studies [14,22,25].

### 3.2. Ni(II)/Pb(II) Adsorption Efficiency

#### 3.2.1. Effect of Initial Concentration

In this study, the initial concentration of Ni(II) and Pb(II) were varied from 10 mg.L^−1^ to 100 mg.L^−1^. As shown in Appendix A, the removal efficiency of Ni(II) and Pb(II) onto H-PBC decreased from 13.61% to 5.34% and 49.86% to 22.51%, respectively. Similarly, with the increase in initial concentration, the removal efficiency of Ni(II) and Pb(II) by other adsorbent also showed a downward trend. The reduction in percentage removal of heavy metals with increasing initial concentration can be attributed to insufficient active sites [13].

#### 3.2.2. Effect of Adsorbent Dosage

Appendix A shows the effect of adsorbent dosage on Ni(II) and Pb(II) removal efficiency. In general, the Ni(II) removal efficiency improved significantly as the amount of adsorbent increased. When the adsorbent dosage was 50 mg, the Ni(II) removal efficiencies using PBC, H-PBC, E-PBC, and O-PBC were 11.2%, 43.1%, 22.6%, and 39.8%, respectively. The efficiency of Ni(II) removal by H-PBC was significantly higher than that by PBC, possibly due to the presence of more binding sites on the surface of H-PBC [3]. Similarly, the concentration of the adsorbent greatly affected the adsorption of Pb(II). The Pb(II) removal efficiency increased rapidly as the adsorbent dosage increased. At a dosage of 20 mg, the Pb(II) removal efficiencies with H-PBC, E-PBC, and O-PBC were 70.0%, 37.5%, and 63.3%, respectively, which were far higher than that with PBC (10.8%). Therefore, the modifications of PBC greatly enhanced the efficiency of heavy metal removal.

#### 3.2.3. Effect of pH

In general, the adsorption process associated with surface functional groups is influenced by the pH of the solution [26]. Zeta potential reflects the charge of the adsorbent surface, negative value of zeta potential corresponding to the negative surface charge of materials [27]. As shown in Appendix A, the isoelectric point of PBC, H-PBC, E-PBC, and O-PBC was about 3.09, 2.55, 3.45, and 3.78, respectively. When pH values were higher than the isoelectric point, the surface of the samples was negatively charged. The Ni(II) removal efficiency improved dramatically as the pH increased from 2 to 9 (Appendix A). This change might have occurred because the functional groups on the surfaces of the adsorbents were negatively charged due to deprotonation as the pH increased, and thus, they combined with Ni(II) via electrostatic attraction, so the removal efficiency increased [15]. When the pH increased to 9, the Ni(II) removal efficiencies with H-PBC, O-PBC, E-PBC, and PBC were 12.8%, 11.8%, 7.6%, and 4.3%, respectively. Moreover, the effect of the pH on Pb(II) was similar to that on Ni(II), where the adsorption capacity improved as the pH increased (Appendix A). When the pH increased to 9, the Pb(II) adsorption efficiencies with H-PBC, O-PBC, E-PBC, and PBC were 42.5%, 41.4%, 25.7%, and 12.1%, respectively.

### 3.3. Thermodynamic Analysis

Temperature is an important parameter in the adsorption process. The effects of temperature on the adsorption of Ni(II) and Pb(II) are shown in Appendix A. The heavy metal removal efficiencies gradually improved as the temperature increased. Moreover, H-PBC had the highest capacity for adsorbing Ni(II) and Pb(II). When the temperature reached 45 °C, the Ni(II) and Pb(II) removal efficiencies with H-PBC were 10.6% and 39.4%, respectively. The thermodynamic behavior was investigated further using Equations (7)–(9) [28]:(7)$$\Delta G^{0}=-RT\ln K_{C}$$
(8)$$K_{C}=\frac{C_{Ae}}{C_{e}}$$
(9)$$\ln K_{C}=-\frac{\Delta G^{0}}{RT}=\frac{\Delta S^{0}}{R}-\frac{\Delta H^{0}}{RT}$$
where $∆G^{0}$ is the Gibbs free energy (kJ/mol), $∆S^{0}$ is the entropy change (J/mol·K), $∆H^{0}$ is the enthalpy change (kJ/mol), *R* is the general gas constant (J/mol·K), *T* is the solution temperature (K), *C_e_* is the concentration of the compound at equilibrium (mg/L), and *C_Ae_* is the amount adsorbed after reaching equilibrium at a specific temperature (mg/g).

The values of $∆H^{0}$ and $∆S^{0}$ were calculated from the plot of ln*K_c_* versus 1/*T* (Appendix A). The thermodynamic parameters are presented in Table 2 and Table 3. The negative values of $∆G^{0}$ for the adsorption of Ni(II) and Pb(II) indicated the spontaneous nature of the adsorption process. The adsorption of heavy metals was more efficient at high temperatures because $∆G^{0}$ decreased as the temperature increased. $∆H^{0}$ was positive in the cases of Ni(II) and Pb(II) adsorption, thereby indicating the endothermic nature of their adsorption. The positive values of $∆S^{0}$ reflected increased randomness on the adsorbents/solution surface and the affinity of Ni(II) and Pb(II) for the adsorbents [23].

### 3.4. Adsorption Kinetics

Pseudo-first order and pseudo-second order models were employed to describe the kinetic adsorption processes. As shown in Figure 3A,B, the pseudo-first order model did not obtain a good linear relationship, but the experimental data were in agreement with the pseudo-second order model. For Ni(II), the correlation coefficients (R^2^ values) obtained for the pseudo-first order models for PBC, H-PBC, E-PBC, and O-PBC were 0.967, 0.960, 0.871, and 0.812 (Table 4), respectively, which were lower than those for the pseudo-second order models (0.994, 0.995, 0.993, and 0.983). The experimental results obtained for Pb(II) were consistent with those for Ni(II). For Pb(II), the correlation coefficients (*R*^2^ values) obtained for the pseudo-first order models for PBC, H-PBC, E-PBC, and O-PBC were 0.883, 0.822, 0.880, and 0.867 (Table 5), respectively, which were lower than those for the pseudo-second order models (0.989, 0.987, 0.993, and 0.993). Furthermore, the values calculated for the pseudo second order model of *q*_e_ were consistent with the experimental q_e,exp_ values based on Ni(II) and Pb(II). The pseudo-second order kinetic constants (*K_2_* values) were small for Ni(II) and Pb(II), thereby indicating that the adsorption rate decreased as the contact time increased and it was proportional to the amount of adsorption sites [4]. These results suggest that the adsorption process was controlled by chemical adsorption rather than physical adsorption.

### 3.5. Adsorption Isotherms

The Langmuir and Freundlich models were used to simulate the adsorption equilibria for heavy metals by the adsorbents. The fitted models and experimental data are shown in Figure 4, and the calculated parameters are presented in Table 6 and Table 7. Langmuir model obtained a high coefficient of determination (*R*^2^ ≥ 0.99) and it was more suitable than the Freundlich model, where it suggested the homogeneous adsorption of Ni(II) and Pb(II). The maximum monolayer adsorption capacities for H-PBC were determined as 64.94 mg/g for Ni(II) and 243.90 mg/g for Pb(II). According to the Freundlich isotherm, the value of 1/*n* < 1 indicated the favorable adsorption of heavy metals. Moreover, the dimensionless separation factor R_L_ was employed to check the favorability of the adsorption process. Based on *R*_L_ = 1/(1 + K_L_C_0_) [29], the Ni(II) and Pb(II) adsorption processes were thermodynamically favorable because the values of R_L_ ranged from 0 to 1.

The comparison of the maximum adsorption capacities of Ni(II) and Pb(II) using previously reported adsorbents are summarized in Table 8 and Table 9, showing that the adsorption capacity of H-PBC was slightly higher than other adsorbents, such as sawdust [30,31], and straw [32,33]. Different adsorbents exhibit different adsorption properties for heavy metals, which may be caused by the variances in raw material composition and preparation methods, resulting in the composition and pore structure of biochar. Consequently, H-PBC is assumed to be a promising adsorption material to remove heavy metals from water.

### 3.6. Adsorption Mechanism

The adsorption mechanisms for toxic metals on modified-biochar involve the comprehensive influence several types of reactions, such as ions exchange, precipitation, complexation, and physical adsorption [40,41]. For H-PBC, the larger S_BET_ value and the experimental data (*R*^2^ > 0.87) fitted well by Freundlich model, indicated that physical adsorption was involved in the process. Moreover, FITR analysis demonstrated the presence of carboxyl and hydroxyl groups, which might facilitate the adsorption process via ion exchange and hydrogen bonding. Jiang et al., [8] synthesized H_3_PO_4_-HC via a hydrothermal carbonization process and found that the oxygen-containing functional groups on H_3_PO_4_-HC, such as C=O groups, played the main role in Pb(II) adsorption. Additionally, the electrostatic interaction may also affect adsorption process, which could prove in adsorption process under different pH conditions. For E-PBC, the main adsorption mechanism may be complexation. Li et al., [15] found that the formation of -NH_2_Pb when a new peak was detected in N spectra after using EDTA-modified biochar to adsorb Pb(II) and attributed the complexation between -NH_2_ and Pb(II). FITR analysis demonstrated the presence of -NH_2_ in E-PBC. For O-PBC, FITR analysis proved the existed of -OH, C-O, which could enhance the adsorption process via precipitation and complexation.

### 3.7. Regeneration Study

The recyclability of the adsorbents is of great importance to evaluate the economic feasibility and propensity of secondary pollution [42]. Figure 5 shows that the adsorption efficiency generally remained unchanged in the first three cycles for H-PBC, E-PBC, O-PBC, and PBC. Furthermore, after three cycles of adsorption and desorption, H-PBC showed good recycling ability with 49.8% and 56.3% for Ni(II) and Pb(II), respectively, thereby indicating that the H-PBC composite material is a stable adsorbent for use in heavy metal removal. However, the adsorption efficiency decreased in the fourth and fifth cycles, probably due to the blockage of some active sites by adsorption and losses of biochar during recovery [43].

## 4. Conclusions

In this study, H-PBC, E-PBC, and O-PBC were successfully prepared and applied for Ni(II) and Pb(II) adsorption. The results obtained in adsorption experiments showed that the efficiencies of Ni(II) and Pb(II) removal were higher with H-PBC, E-PBC, and O-PBC than PBC. H-PBC has the maximum adsorption capacities for Ni(II) and Pb(II) was 64.94 mg/g and 243.90 mg/g, respectively. The adsorption kinetics and the adsorption isotherms fitted a pseudo-second order model and the Langmuir adsorption model, respectively. The stable regeneration of H-PBC (49.8% and 56.3% for Ni(II) and Pb(II)) was verified by three consecutive adsorption-desorption cycles. These results demonstrate that the excellent adsorption performance of H-PBC to heavy metals is a possible option for wastewater treatment.

## Figures and Tables

**Figure 1 ijerph-19-11163-f001:**
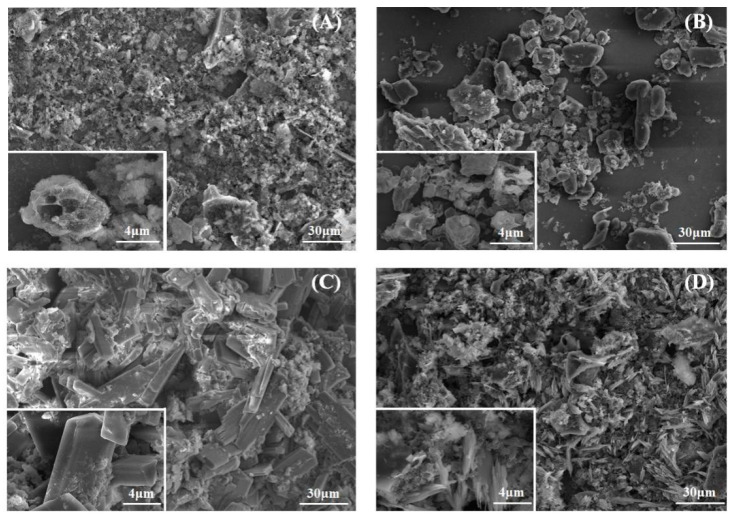
Scanning electron microscopy images of (**A**) PBC; (**B**) E-PBC; (**C**) O-PBC, and (**D**) H-PBC.

**Figure 2 ijerph-19-11163-f002:**
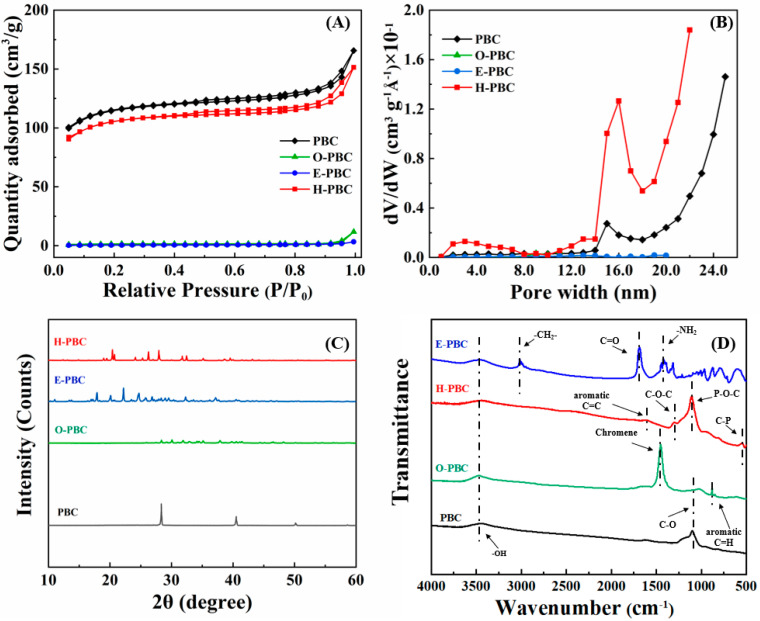
(**A**) Brunauer-Emmett-Teller spectra, (**B**) pore size distributions, (**C**) X-ray diffraction patterns, and (**D**) Fourier-transform infrared spectra obtained for PBC, E-PBC, O-PBC, and H-PBC.

**Figure 3 ijerph-19-11163-f003:**
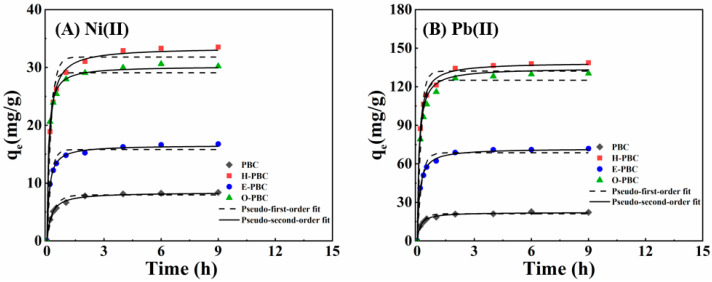
Adsorption kinetic equations fitted for (**A**) Ni(II) and (**B**) Pb(II) with PBC, H-PBC, E-PBC, and O-PBC. Experimental conditions: adsorbent dosage = 10 mg; [Ni(II)] and [Pb(II)] = 40 mg/L; pH = 5.0; temperature = 298.15 K.

**Figure 4 ijerph-19-11163-f004:**
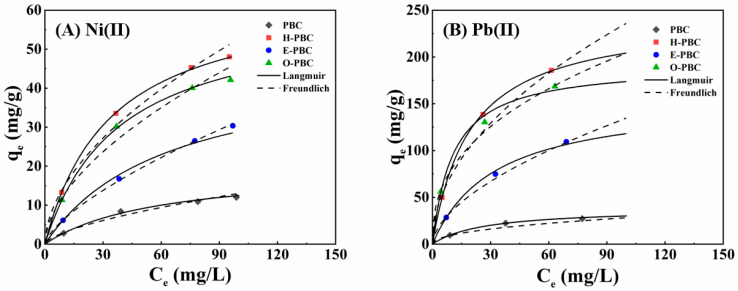
Sorption isotherms for (**A**) Ni(II) and (**B**) Pb(II) with PBC, H-PBC, E-PBC, and O-PBC. Experimental conditions: adsorbent dosage = 10 mg; pH = 5.0; temperature = 298.15 K; adsorption time = 9 h.

**Figure 5 ijerph-19-11163-f005:**
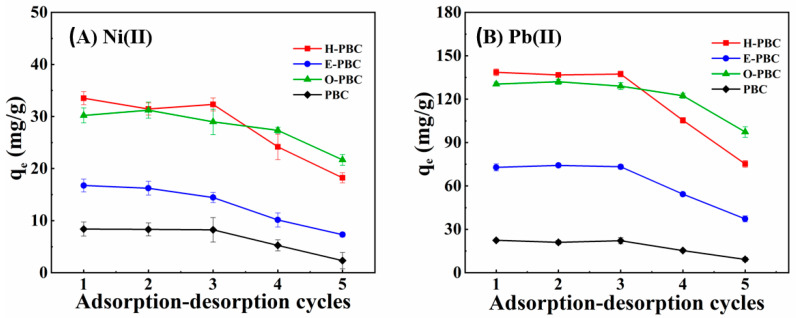
Recyclability analysis for the adsorption of (**A**) Ni(II) and (**B**) Pb(II) by PBC, H-PBC, E-PBC, and O-PBC. Experimental conditions: adsorbent dosage = 200 mg; [Ni(II)] and [Pb(II)] = 40 mg/L; adsorption time = 9 h.

**Table 1 ijerph-19-11163-t001:** Element and porosity characteristic of PBC, H-PBC, O-PBC, and E-PBC.

Samples	Elemental Analysis/%				Element Atomic Ratio				Porosity Characteristic		
	C	H	O	N	P	O/C	H/C	(O+N)/C	BET Surface Area (m^2^.g^−1^)	BJH Pore Volume (cm^3^.g^−1^)	Average Particle Size (Å)
PBC	61.40	2.95	14.76	1.52	-	0.18	0.58	0.20	360.41	0.213	5.89
H-PBC	64.33	2.42	23.16	0.37	7.64	0.27	0.45	0.28	344.17	0.234	2.87
E-PBC	58.64	1.32	9.48	3.66	-	0.12	0.27	0.17	1.64	0.005	2.47
O-PBC	60.98	0.77	8.97	0.68	-	0.11	0.15	0.12	3.66	0.002	1.91

**Table 2 ijerph-19-11163-t002:** Thermodynamic parameters for the adsorption of Ni(II) by PBC, H-PCB, E-PBC, and O-PBC at different temperatures.

Adsorbate	Adsorbent	∆*H* (kJ mol^−1^)	∆*S* (J mol^−1^ K^−1^)	∆*G* (kJ mol^−1^)			
				288.15 K	298.15 K	308.15 K	318.1 K
Ni(II)	PBC	6.79	23.6	−0.02	−0.25	−0.48	−0.72
	H-PBC	15.15	57.00	−1.27	−1.84	−2.41	−2.98
	E-PBC	15.98	56.00	−0.16	−0.72	−1.27	−1.83
	O-PBC	12.23	46.00	−1.02	−1.48	−1.94	−2.40

**Table 3 ijerph-19-11163-t003:** Thermodynamic parameters for the adsorption of Pb(II) by PBC, H-PCB, E-PBC, and O-PBC at different temperatures.

Adsorbate	Adsorbent	∆*H* (kJ mol^−1^)	∆S (J mol^−1^ K^−1^)	∆*G* (kJ mol^−1^)			
				288.15 K	298.15 K	308.15 K	318.1 K
Pb(II)	PBC	8.62	31.00	−0.31	−0.62	−0.93	−1.24
	H-PCB	4.27	32.00	−4.95	−5.27	−5.59	−5.91
	E-PBC	5.85	34.00	−3.94	−4.28	−4.62	−4.96
	O-PBC	4.68	33.00	−4.83	−5.16	−5.49	−5.82

**Table 4 ijerph-19-11163-t004:** Kinetic parameters for pseudo-first order and pseudo-second order models of Ni(II) adsorption by PBC, H-PBC, E-PBC, and O-PBC.

Samples	*Q*e, exp (mg/g)	Pseudo First-Order			Pseudo Second-Order		
		*q*_e1_ (mg/g)	*K*_1_ (h^−1^)	*R* ^2^	*q*_e2_ (mg/g)	*K*_2_ (mg.(gh)^−1^)	*R* ^2^
PBC	8.39	7.95	2.91	0.967	8.40	0.57	0.994
H-PBC	33.52	31.79	4.54	0.960	33.44	0.24	0.995
E-PBC	16.75	15.80	4.95	0.871	16.56	0.54	0.993
O-PBC	30.21	29.08	6.36	0.812	30.21	0.42	0.983

**Table 5 ijerph-19-11163-t005:** Kinetic parameters for pseudo-first order and pseudo-second order models of Pb(II) adsorption by PBC, H-PBC, E-PBC, and O-PBC.

Samples	*Q*_e_, exp (mg/g)	Pseudo First-Order			Pseudo Second-Order		
		*q*_e1_ (mg/g)	*K*_1_ (h^−1^)	*R* ^2^	*q*_e2_ (mg/g)	*K*_2_ (mg.(gh)^−1^)	*R* ^2^
PBC	22.21	21.13	3.85	0.883	22.22	0.293	0.989
H-PBC	138.57	132.14	5.57	0.822	138.89	0.077	0.987
E-PBC	71.86	68.62	4.53	0.880	71.94	0.113	0.993
O-PBC	130.47	125.05	5.03	0.867	137.58	0.072	0.993

**Table 6 ijerph-19-11163-t006:** Parameters fitted to Langmuir and Freundlich models of Ni(II) adsorption by PBC, H-PBC, E-PBC, and O-PBC.

	Langmuir			Freundlich		
	*q*_m_ (mg/g)	*K*_L_ (L.mg^−1^)	*R* ^2^	*K* _F_	1/*n*	*R* ^2^
PBC	19.80	0.017	0.998	0.704	0.633	0.875
H-PBC	64.94	0.0295	0.996	4.225	0.548	0.885
E-PBC	47.17	0.0158	0.998	1.323	0.669	0.995
O-PBC	60.24	0.0262	0.999	3.493	0.563	0.978

**Table 7 ijerph-19-11163-t007:** Parameters fitted to Langmuir and Freundlich models of Pb(II) adsorption by PBC, H-PBC, E-PBC, and O-PBC.

	Langmuir			Freundlich		
	*q*_m_ (mg/g)	*K*_L_ (L.mg^−1^)	*R* ^2^	*K* _F_	1/*n*	*R* ^2^
PBC	38.31	0.037	0.999	3.368	0.491	0.981
H-PBC	243.90	0.0514	0.999	23.58	0.500	0.875
E-PBC	156.25	0.031	0.968	9.34	0.579	0.975
O-PBC	192.31	0.0935	0.997	31.907	0.403	0.939

**Table 8 ijerph-19-11163-t008:** Comparison of various representative Ni(II) adsorbents.

Years	Biomass Feedstock	Finishing Materials	Maximum Adsorption Capacity (mg.g^−1^)	Reference
2017	Straw		25.06	[33]
	Rice husk		10.15	
2013	Figs	H_3_PO_4_	18.78	[34]
2017	Sludge	Mercaptan	52.40	[35]
2017	Sawdust	KOH	94.49	[31]
		ZnCl_2_	19.36	
2020	Cactus	NaOH	44.35	[11]
2020	Rice bran	Al(NO_3_)_3_, Mg(NO_3_)_2_, FeSO_4_	201.62	[36]
	Corn		19.80	This study
		H_3_PO_4_	64.94	
		EDTA	47.17	
		NaOH	60.24	

**Table 9 ijerph-19-11163-t009:** Comparison of various representative Pb(II) adsorbents.

Years	Biomass Feedstock	Finishing Materials	Maximum Adsorption Capacity (mg.g^−1^)	Reference
2014	Eggshell	ALG and PEI	344.8	[37]
2014	Bean shell		45.3	[38]
2014	Sawdust	H_3_PO_4_	80.65	[30]
2017	Banana peel	Ammonium sulfate persulfate	315.16	[39]
2018	Straw		134.68	[32]
2019	Shell	EDTA	129.31	[15]
2019	Peanut shells	H_3_PO_4_	353.4	[14]
		PEI	214.0	
2019	Lignin	PEI and CS_2_	79.9	[39]
	Corn		38.31	This study
		H_3_PO_4_	243.90	
		EDTA	156.25	
		NaOH	192.31	

## Data Availability

The data and materials used and analyzed during the current study are available from the corresponding author on reasonable request.

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
