# Peer review of "Comparative Study of Biochar Modified with Different Functional Groups for Efficient Removal of Pb(II) and Ni(II)"

_ijerph, 2022, doi:10.3390/ijerph191811163_

Round 1
Reviewer 1 Report
|
Section |
Review comments and notes |
|
Abstract, title and references |
-The manuscript title is relevant and sufficient and enough informative. -The authors are well presented the study background and aims, methods, results and conclusion in the abstract. -Line no. 25. Can you give a % of Regeneration analysis showed that H-25 PBC had superior reusability characteristics? -The list of references is relevant to this study. -The authors are request to add some reference from recent of top journals. -The list of reference is short, need add more reference. -the keyword should be revised |
|
Introduction/ background |
-Line no. 33-34. The pollution of aquatic environment by heavy metals maybe both natural and anthropogenic. Should be corrected -It would be better if figure out both Ni and Pb impact on human and environment in the developed and developing countries. -Last decade extensive research article has been published about biochar and its application in water and wastewater treatment. Resulting, the text is common and seems plagiarize. However, the introduction part is well presented everything, just need to little bit revised some section, add more relevant citation in the innovation section, what did other author section also. -Research gap need to be more strong as we know last decade huge article has been published in this field. -Can you mention a sentence, why your method is important? |
|
Methods |
-Materials and reagent section is missing -Line no. 97. Add heating rate. -Line no. 109. Pls. revised the sentence. Dried in oven. -Line no. 102. Should mention reactor specification -Line no. 114; Pls. add specification of the SEM, XRD, FTIR, and BET. -Line no. 200. Need a ref. -Line no. 177. The surface area of O-PBC was also significantly reduced. The reason is similar to E-PBC |
|
Results & Discussion |
-Line no. 307; why the adsorption efficiency generally remained unchanged in the first three cycle? -What will be the ultimate fate of the adsorbent after 5th recycle? -The authors are request to add a table on “compare to the other studies”. For details pls. click the link 10.22104/AET.2022.5087.1379 -Adsorption mechanism is missing. For details pls. click the link 10.22104/AET.2022.5087.1379 - For Regeneration study, section, the author may follow the article, https://doi.org/10.1016/j.apcatb.2019.117765 -Mechanism of Ni(II) and Pb(II) adsorption, is missing. -in the result and discussion section, we expected a comparative study of adsorption capacity of various adsorbent. |
|
Conclusion |
-the conclusion section focused key findings of the study -Can you mention the regeneration capacity in the conclusion part? As it is express the sustainability of the innovation. |
|
Overall |
I read the paper carefully and I wish to support it for publication after major revision. This is an original work and depth of research is accurate. The manuscript has been written in Standard English. The research question is well defined. In my eyes, the main comments of the paper are as follows:
1) Materials and reagent section is missing 2) Mechanism of Ni(II) and Pb(II) adsorption, is missing. 3) adsorption capacity of various adsorbent, is missing 4) Research gap need to be more strong as we know last decade huge article has been published in this field. Overall, the paper should need major revision. |

Author Response
Many thanks to the reviewer for the recognition and support. Your constructive suggestions are of great help in improving our manuscript. We have carefully revised the manuscript according to the suggestions. All the changes were marked with red color in the revised manuscript.

Reviewer 2 Report
The manuscript “Comparative study of biochar modified with different functional groups for efficient removal of Pb(II) and Ni(II)” is describing the modified biochar made from cornstalk and their applications for removal of heavy metals removal from aqueous solution. Generally, it is a textbook-like adsorption work, so I can image that it contains simple material characterization, effect of solution chemistry factors, isotherms, and kinetics. Having read the whole paper carefully, my concern is that I could not find something new in this work. There is a considerable lack of explanation of mechanisms; thus, more detail discussion should be presented in the section of results and discussion, as well as mechanism enhancement by biochar, to provide more credibility for the findings. In addition, there are some typos in whole manuscript. The authors need to carefully check its English and typo errors before its submission to a journal. Therefore, it is necessary to improve the quality of this manuscript to be published in this journal.
Additional comments
· The language quality of manuscript needs to be improved significantly.
· Material name used for biochar production should be included in abstract as well. Abbreviations of biochar are quite confusing and needs to be revised e.g. H3PO4-modified biochar (H-PBC); H stands for first word but for modified biochar the term PBC is not matching. Moreover, abbrivations for Brunauer–Emmett–Teller analysis should be presented i.e BET analysis like other FITR and SEM. all exhibited excellent performance at (it should be for not at) Ni(II) and Pb(II).
· Add surface areas of biochars in abstract for proper linkage to high adsorption capacity of materials
· L36 delete , before reference. Replace Ni with Ni(II), wherever it applied in the text. Please do it for other metals as well
· What was the selection criteria for heavy metals? Why only Ni(II) and Pb (II) were selected for removal.
· Although, sufficient literature is available and work has been done. Please elaborate the novelty of this work?
· Some terms like porogen, Pristine etc. are not common and should be avoided. Please replace all these words with some suitable one.
· The modified biochars were applied for Ni(II) and Pb (II) removal and their removal efficiency was assessed. Why biochars were not prepared at the same condition? As for my understanding it is more suitable and justifying to prepare the materials at same condition and then assessing the efficiencies by applying for pollutant removal. In its present form it does not make any logic and reflecting the poor experimental design.
· Elemental analysis of feedstock and biochars is necessary to be performed to assess the C, H and O contents and their relation is further linked with char quality for better understanding the removal mechanism.
· Please avoid the terms like home made reactor. Either it could be laboratory scale reactor etc.
· L121 is not grammatically correct. Biochar is mixed with solution instead of placing in reactor.
· What was time interval? As it is important to determine the equilibrium first.
· Give model of centrifuge please
· Add references for equations used.
· The quality of figures is poor and needs to be improved. In its present form it is hard to read the figures.
· I think the data presented in S-files should be part of main document. As pore volume data, thermodynamics data and models data is important in view of presented discuss in.
· Where is effect of time and concentration (soln) data? The calculation of Qmax and models fitting relies well o the conc varying effects. Please justify.
· More discussion required for results and discussion
· Conclusion needs to be revised.
Author Response

(The authors gave the same response as above.)

Reviewer 3 Report
Liu et al. studied the properties of corn biochar modified with different functional groups for the removal of Ni(II) and Pb (II). They found that phosphorous containing biochar is the most efficient one for the removal of Ni (II) and Pb (II), demonstrating that the modification of porous biochar is a very effective tool for the removal of heavy metals
In my opinion the paper has the quality to be published in International Journal of Environmental research and Public health.
I would suggest just minor modifications
PARAGRAPH 3.1
the SEM pictures could be added to the main text
Question: Why the hysteresis loop is defined as H4 and not H3?
Paragraph 3.2.1
The removal efficiency should be defined: how did you calculate it?
In fig S2(A) you could keep the same colour of figure 2 S(B) to increase readability
Paragraph 3.4
Is the term qe2 referred to the qe of the pseudo second order model?
Author Response

(The authors gave the same response as above.)

Reviewer 4 Report
Reviewer comments:
This manuscript (ijerph-1865042) studied the comparative study of biochar modified with different functional groups for efficient removal of Pb(II) and Ni(II). The research is not novel and informative, meanwhile the following problems need to be addressed.
1. In the Introduction Section, why do authors choose NaOH, EDTA, H3PO4 as modifiers, please explain in detail. Moreover. What are the functional groups represented by H-PBC, E-PBC, O-PBC?
2. Page 1 lines 75-76, “The most suitable biochar modification method to enhance the removal of both Pb and Ni is still not clear”, the expression of sentence is inaccurate. Please polish the main text of the manuscript.
3. Page 2, lines 92-94, It can only guarantee that the particle size is less than 150 μm, but the average particle size cannot be determined, and the value of the particle size cannot be guaranteed to be constant.
4. Page 3, lines 96-98, Biochar refers to a solid material obtained by thermochemical conversion of biomass in an oxygen-free or oxygen-limited environment, and the temperature range of pyrolysis is (3-700°C). Although previous studies have used temperatures higher than 700°C, an explanation needs to be provided as to whether the obtained biochar can be called biochar at 800°C. At 800°C, the number of biochar functional groups is low, the pores collapse, and the overall adsorption performance will decrease. Why choose this temperature? Also, please explain the specific purpose of cleaning with hydrochloric acid.
5. Page 3 lines 100-113, why not choose the same synthesis conditions to prepare H-PBC, E-PBC, O-PBC. How H-PBC, O-PBC, E-PBC prepared by different modifiers using different synthesis methods prove that H-PBC is the optimal adsorbent compared to other adsorbents. Do the different synthesis processes affect your conclusions?
6. Page 5, Fig1 C and D are difficult to identify.
7. Page 5, Section 3.2.2, It is recommended to supplement zeta isoelectric point and solubility product for a more adequate explanation.
8. Page 6, lines 232-233, When the experimental temperature is 50 to 60 °C, the solution will evaporate a lot, which will affect the accuracy of the experiment. Please adjust the temperature range appropriately.
Author Response

(The authors gave the same response as above.)

Round 2
Reviewer 1 Report
-What will be the ultimate fate of the adsorbent after 5th recycle? This question couldn’t address properly. What the fate of adsorbent after 5 recycle? Is it environment friendly?
-The authors are request to add a table on “compare to the other studies”. For details pls. click the link 10.22104/AET.2022.5087.1379. the authors are inserted in Table supplementary section. It’s alright but need a discussion and comparison in the manuscript with citation. The author may click and cite the mentioned link.
- Research gap need to be more strong as we know last decade huge article has been published in this field. The author should careful about this section and revised it.

Author Response

(The authors gave the same response as above.)

Reviewer 2 Report
The authors have made significant efforts to improve the manuscript according to the suggestions. Prior to acceptance, few corrections needs to be done as given below
1. Materials: Lead acetate formula. Please use subscript for the numbers
2. Eq.2: Removal efficiency (%)=equation and remove % sign
3. Why only langmuir and Freundlich models were used for isotherm. There are other models i.e. Temkin etc to be evaluated.
4. Fig. 2 (D). Please recheck the title of Y-axis. I think it should be Transmittance instead of adsorbance.
5. Influence of conc. variation on removal % is important to determine. Please provide the effect of concentration variation impact on removal efficiency of heavy metals.
6. Please add the details for XRD and FTIR analysis relation to selected heavy metals removal. From presented information i can just only see the description of graph itslef instead of their role describing for abatement.
7. Please change Ni(II) and Pb (II) in Figure 3 and 4.
Author Response

(The authors gave the same response as above.)

Reviewer 4 Report
The manuscript has not been improved significantly. The manuscript has not been sufficiently improved to warrant publication in IJERPH.
Author Response

(The authors gave the same response as above.)
